

# Sivers extraction with Neural Network

**I. P. Fernando, N. Newton, D. Seay and D. Keller**

Department of Physics, University of Virginia, Charlottesville, VA 22904, USA

⋆ ishara@virginia.edu

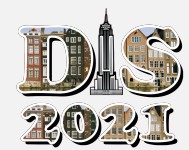

*Proceedings for the XXVIII International Workshop
on Deep-Inelastic Scattering and Related Subjects,
Stony Brook University, New York, USA, 12-16 April 2021*

## Abstract

**Pseudo-data with simulated experimental errors can be generated to train an ensemble of Artificial Neural Networks (ANN) implemented on a regression to extract Transverse Momentum-dependent Distributions (TMDs). A preliminary analysis is presented on the reliability in extraction of the Sivers function imposed in the pseudo-data given the bounds on the experimental errors, data sparsity, and complexity of phase-space.**

## 1 Introduction

Transverse Momentum Dependent Parton Distribution Functions (TMD PDFs) can be extracted from the processes that are corresponding to multiple kinematic scales such as Drell-Yan (DY), Semi Inclusive Deep Inelastic Scattering (SIDIS), and $e^+ e^-$ annihilation. Therefore, the cross-sections (or differential cross-sections) measured from these processes are sensitive to the transverse momentum of partons, especially the magnitude of that momentum corresponding to non-negligible non-perturbative interactions. The original proposal for TMD PDFs was introduced by Collins, Soper, and Sterman [1–3]. PDFs provide $f(x)$ the parton density in terms of light-cone momentum fraction $(x)$, whereas TMD PDFs provide $f(x, \mathbf{k}_\perp)$ the parton density as a function of both light-cone momentum fraction and transverse momentum. There are eight TMD PDFs at the leading-twist, or in other words twist-2 approximation: $\mathcal{O}\left(1/Q^2\right)$, which can be classified in terms of quark-polarization and nucleon-polarization (see Figure (1)). Among those eight TMD PDFs, there are two time-reversal odd TMDs, namely Sivers function & Boer-Mulders function, which represent the correlation between the spin of the quark and the spin of the hadron. The Sivers function corresponds to the polarized hadron, and the Boer-Mulders function corresponds to the unpolarized hadron.

Sivers [5, 6] suggested that the $k_\perp$ distribution could have an azimuthal asymmetry when the initial hadron is transversely polarized, but this is in contradiction only with "Parity" and "Time-reversal" invariance (PT) of QCD. In other words, this asymmetry does not exist according to the PT invariance of QCD. The Sivers function is the correlation between unpolarized quarks in a transversely polarized nucleon. It vanishes by its naive definition [7].

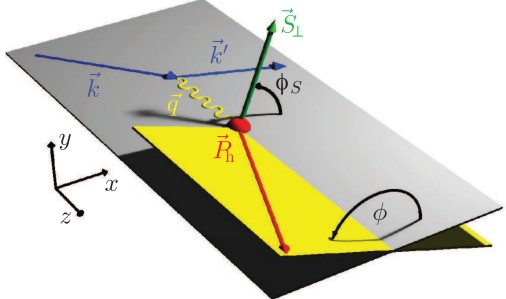

Figure 1: Left: Leading twist TMDs categorized according to the polarization of the quarks and the nucleons. Right: Semi-inclusive hadron production in DIS processes [4]

## 2   Sivers asymmetry from SIDIS

In Semi Inclusive Deep Inelastic Scattering (SIDIS) process, the differential cross-section depends on both collinear parton distribution functions $f_{q/p}(x)$ and fragmentation functions $D_{h/q}(z)$, where $q$ is the quark flavor, $p$ represents the target proton, $h$ is the hadron type produced by the process and $z$ is the momentum fraction of the final state hadron with respect to the virtual photon. A simplified version of the SIDIS differential cross-section can be written as,

$$\frac{d^5\sigma^{lp\to lhX}}{dxdQ^2dzd^2p_{hT}} \propto \sum_q e_q^2 \int d^2\mathbf{k}_\perp \, \mathcal{K}(x,p_{hT},Q^2)\hat{f}_{q/p\uparrow}(x,\mathbf{k}_\perp)D_{h/q}(z,p_\perp) + \mathcal{O}(\mathbf{k}_\perp/Q) , \quad (1)$$

where $\mathcal{K}(x,p_{hT},Q^2)$ represents the factorized kinematical factor. $\hat{f}_q(x,\mathbf{k}_\perp)$ is the unpolarized quark distribution with transverse momentum $\mathbf{k}_\perp$ inside a transversely polarized (with spin $\mathbf{S}$) proton with 3-momentum:

$$\hat{f}_{q/p\uparrow}(x,\mathbf{k}_\perp) = f_{q/p}(x,k_\perp) + \frac{1}{2}\Delta^N f_{q/p\uparrow}(x,k_\perp)\mathbf{S}.(\hat{\mathbf{p}}\times\hat{\mathbf{k}}_\perp) \quad (2)$$

and $\Delta^N f_{q/p\uparrow}(x,k_\perp)$ is the Sivers function which contains the effect on quarks due to the spin-polarization of the proton. The Sivers asymmetry in SIDIS process (see Fig.[1]) can be written in terms of the cross-sections as,

$$A_{UT}^{\sin(\phi_h-\phi_S)}(x,y,z,p_{hT}) = \frac{d\sigma^{l\uparrow p\to hlX} - d\sigma^{l\downarrow p\to lhX}}{d\sigma^{l\uparrow p\to hlX} + d\sigma^{l\downarrow p\to hlX}} \equiv \frac{d\sigma\uparrow - d\sigma\downarrow}{d\sigma\uparrow + d\sigma\downarrow}, \quad (3)$$

which can be parameterized as [8],

$$A_{UT}^{\sin(\phi_h-\phi_S)}(x,y,z,p_{hT}) = \frac{[z^2\langle k_\perp^2\rangle + \langle p_\perp^2\rangle]\langle k_S^2\rangle^2}{[z^2\langle k_S^2\rangle + \langle p_\perp^2\rangle]\langle k_\perp^2\rangle^2} \exp\left[-\frac{p_{hT}^2\, z^2\left(\langle k_S^2\rangle - \langle k_\perp^2\rangle\right)}{\left(z^2\langle k_S^2\rangle + \langle p_\perp^2\rangle\right)\left(z^2\langle k_\perp^2\rangle + \langle p_\perp^2\rangle\right)}\right]$$
$$\times \frac{\sqrt{2e}\, z\, p_{hT}}{M_1}\frac{\sum_q \mathcal{N}_q e_q^2 f_q(x)D_{h/q}(z)}{\sum_q e_q^2 f_q(x)D_{h/q}(z)}, \quad (4)$$

where, $\phi_S, \phi_h$ are azimuthal angles of the final state hadron and the transverse polarization vector of the nucleon with respect to the lepton plane; and $P_{hT}$ is the transverse momentum

of the final state hadron with respect to the virtual photon in the center-of-mass of the virtual photon and the nucleon. The TMD Fragmentation Functions are parameterized as,

$$D_{h/q}(z, p_\perp) = D_{h/q}(z)\frac{1}{\pi\langle p_\perp^2\rangle}e^{-p_\perp^2/\langle p_\perp^2\rangle}, \tag{5}$$

where $D_{h/q}(z)$ is the fragmentation function for a quark with flavor $q$ in a hadron type $h$. $\langle k_S^2\rangle = \frac{M_1^2\langle k_\perp^2\rangle}{M_1^2+\langle k_\perp^2\rangle}$; and $\langle k_\perp^2\rangle$, $\langle p_\perp^2\rangle$ were fixed to $0.57 \pm 0.08$ GeV$^2$, and $0.12 \pm 0.01$ GeV$^2$ as in [8] by fitting to the multiplicities from HERMES data. Assuming the Gaussian factorized form [8,9], the unpolarized TMDs and the Sivers function can be parameterized as,

$$f_{q/p}(x, k_\perp) = f_q(x)\frac{1}{\pi\langle k_\perp^2\rangle}e^{-k_\perp^2/\langle k_\perp^2\rangle}, \tag{6}$$

$$\Delta^N f_{q/p^\uparrow} = 2\mathcal{N}_q(x)h(k_\perp)f_{q/p}(x, k_\perp), \tag{7}$$

where,

$$\mathcal{N} = N_q x^{\alpha_q}(1-x)^{\beta_q}\frac{(\alpha_q+\beta_q)^{(\alpha_q+\beta_q)}}{\alpha_q^{\alpha_q}\beta_q^{\beta_q}}, \tag{8}$$

$$h(k_\perp) = \sqrt{2e}\frac{k_\perp}{M_1}e^{-k_\perp^2/M_1^2}. \tag{9}$$

Therefore, one can simplify $A_{UT}^{\sin(\phi_h-\phi_S)}(x, y, z, p_{hT})$ as follows,

$$A_{UT}^{\sin(\phi_h-\phi_S)}(x, y, z, p_{hT}) = \mathcal{A}_0(z, p_{hT}, M_1)\left(\frac{\sum_q \mathcal{N}_q(x)e_q^2 f_q(x)D_{h/q}(z)}{\sum_q e_q^2 f_q(x)D_{h/q}(z)}\right), \tag{10}$$

where

$$\mathcal{A}_0(z, p_{hT}, M_1) = \frac{[z^2\langle k_\perp^2\rangle + \langle p_\perp^2\rangle]\langle k_S^2\rangle^2}{[z^2\langle k_S^2\rangle + \langle p_\perp^2\rangle]^2\langle k_\perp^2\rangle}\exp\left[-\frac{p_{hT}^2 z^2\left(\langle k_S^2\rangle - \langle k_\perp^2\rangle\right)}{\left(z^2\langle k_S^2\rangle + \langle p_\perp^2\rangle\right)\left(z^2\langle k_\perp^2\rangle + \langle p_\perp^2\rangle\right)}\right] \times \frac{\sqrt{2e}z p_{hT}}{M_1}. \tag{11}$$

$f_q(x)$ is the co-linear parton distribution function for flavor $q$ that was obtained from CTEQ6l LHAPDF [10], whereas the fragmentation functions for $\pi^{\pm,0}, K^\pm$ are also extracted from NNFF10_nlo grids which are also available on LHAPDF [11]. There are 13 fitting parameters: $M_1, N_q, \alpha_q, \beta_q, N_{\bar{q}}$, where $q = u, d, s$ and $\bar{q} = \bar{u}, \bar{d}, \bar{s}$; and the fitting routine is *iminuit* (python version of MINUIT) [12]. Two main differences of this work compared to [8] are: (1) LHAPDF grids for FFs (NNFF10_nlo) were used instead of the DSS implementation, (2) $s$ and $\bar{s}$ quark-flavors were considered. Sivers asymmetries data on the SIDIS process, from HERMES (2009) [13] and (2020) [14] were used in this analysis, with the plan of extending the effort towards including all available data from different experiments. Furthermore, a step-forward has been taken to remove the model dependence behavior of $\mathcal{N}_q(x)$ using Neural Network approach.

## 3  Modeling quark contribution with Neural Network

The lack of a satisfactory formulation for the quark contribution leads us to desire an unbiased, trainable model for this aspect of the Sivers asymmetry equation. Thus, in this work $N_q(x)$

is being considered in a model-independent fashion. Neural networks are known to be able to adapt any arbitrary function through the universal approximation theorem [15], and thus are ideal for this application. In this study, we used multilayer feed-forward neural network models to approximate the quark contribution.

These in general operate by receiving inputs to the network, in this case, the kinematic variable $x$. These variables and a trainable matrix of "weights" are convoluted to comprise a hidden layer. To enable the network to approximate non-linear functions, a non-linear activation is applied to the outputs of this intermediate layer. Then these outputs are passed to the next hidden layer, and the process is repeated for an arbitrary number of layers. For the last step of this "forward pass," the outputs of the penultimate layer are passed to the output layer, and the estimate for the target value is obtained. No activation function is applied to the final layer because this is a regression problem.

In training the network, the "backward pass" is used to update the weights of the network with a loss function, which is a measure of how well the network fits the data. The partial derivative of the loss function with respect to each weight in the network is computed, then the weights are adjusted to reduce the magnitude of the loss function. In this problem, we have no direct experimental values for the quark contribution at different values of $x$. Thus, we propagate the outputs of the network through a computational graph (displayed in fig. 2) that computes the entire Sivers function. Then, the weights for the neural network for each quark flavor are updated using the experimental observations of Sivers asymmetries in different kinematic ranges.

The inputs of the computational graph are $x, z,$ and $p_{hT}$. The PDFs and FFs are generated for the corresponding kinematic values, using LHAPDF outside the computational graph. They are then taken as inputs to the graph along with the kinematics. Specifically, $e_q^2 f_q(x) D_{h/q}(z)$ is calculated for each kinematic setting and fed as an input to the computational graph. It is necessary to calculate these expressions independently of the graph because neither the PDFs nor the FFs as taken directly from LHAPDF are automatically differentiable with TensorFlow. These inputs are then passed through the Sivers function as usual, except the neural networks which model the quark contribution for each flavor. These neural networks are each comprised of two dense layers with 32 nodes and ReLU activations. There is a separate network for each quark flavor.

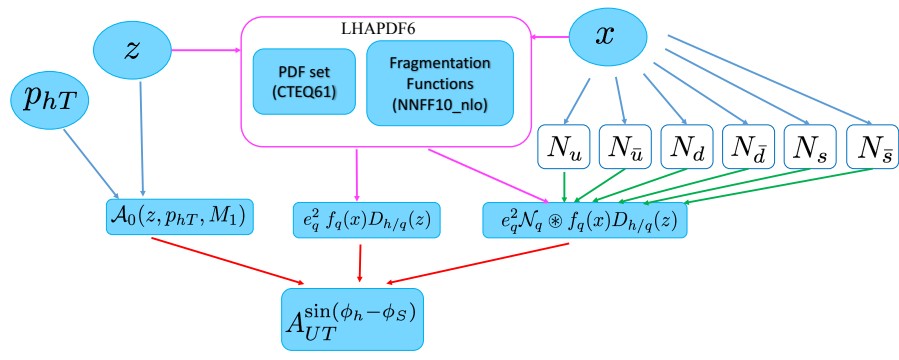

Figure 2: Entire TensorFlow computational graph with kinematics as inputs and Sivers asymmetry as output. Each $N_q$ is a neural network model.

## 4   Results

The fits to the kinematic sets have been performed using *iminuit* (see Table [1]). The neural net has been trained using the HERMES2009 data set. The asymmetry plots HERMES2020 for the NN are predictions which have been compared with the real data. Table [2] summarizes the $\chi^2/dof$'s to compare how well the Sivers asymmetries (with respect to $x, z, P_{hT}$) are described by *iminuit* fits as well as by the Neural Network.

Table 1: Individual fit (*iminuit*) results with HERMES2009 & HERMES2020 data. Note: The acceptance in HERMES2020 in $x$ was $0.023 < x < 0.6$ whereas the corresponding range in HEREMES2009 was $0.023 < x < 0.4$. Also, in HEREMES2020 three-dimensional kinematic binning was considered compared to one-dimensional kinematic binning in either $x, z, or P_{h\perp}$ in HERMES2009.

| Parameter | HERMES 2009 | HERMES2020 |
|:---:|:---:|:---:|
| $M_1$ | $1.303 \pm 0.010$ | $7.590 \pm 0.008$ |
| $N_u$ | $0.169 \pm 0.002$ | $0.960 \pm 0.084$ |
| $\alpha_u$ | $0.645 \pm 0.125$ | $2.291 \pm 0.200$ |
| $\beta_u$ | $3.122 \pm 2.661$ | $9.826 \pm 1.556$ |
| $N_{\bar{u}}$ | $0.007 \pm 0.003$ | $0.205 \pm 0.02$ |
| $N_d$ | $-0.434 \pm 0.005$ | $-4.713 \pm 0.004$ |
| $\alpha_d$ | $1.777 \pm 0.909$ | $0.482 \pm 0.866$ |
| $\beta_d$ | $7.788 \pm 2.144$ | $(5.675 \pm 6.45) \times 10^{-6}$ |
| $N_{\bar{d}}$ | $-0.142 \pm 0.048$ | $1.490 \pm 0.05$ |
| $N_s$ | $0.563 \pm 0.073$ | $4.528 \pm 0.073$ |
| $\alpha_s$ | $(6.84 \pm 10.00) \times 10^{-5}$ | $(1.745 \pm 9.20) \times 10^{-5}$ |
| $\beta_s$ | $(5.987 \pm 8.77) \times 10^{-10}$ | $(6.082 \pm 9.55) \times 10^{-10}$ |
| $N_{\bar{s}}$ | $-0.122 \pm 0.504$ | $8.692 \pm 0.46$ |

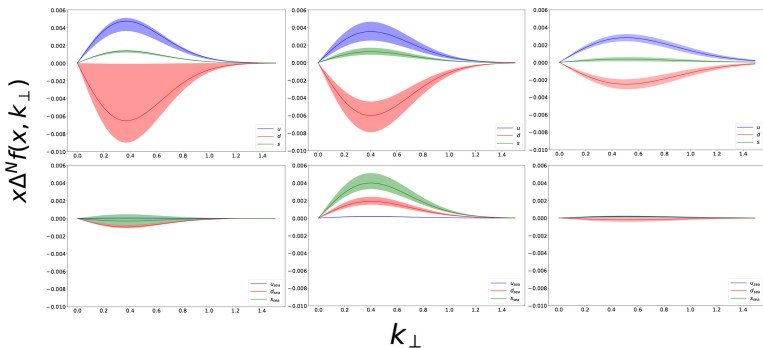

Figure 3: The extracted Sivers functions (at $Q^2 = 2.4$ GeV$^2$) for the valence & sea quarks in the case of $SU(3)_{flavor}$ by *iminuit* fits (first two columns: for HERMES2009 & HERMES2020 accordingly, and the third column: Neural Net fit to HERMES2009 data).

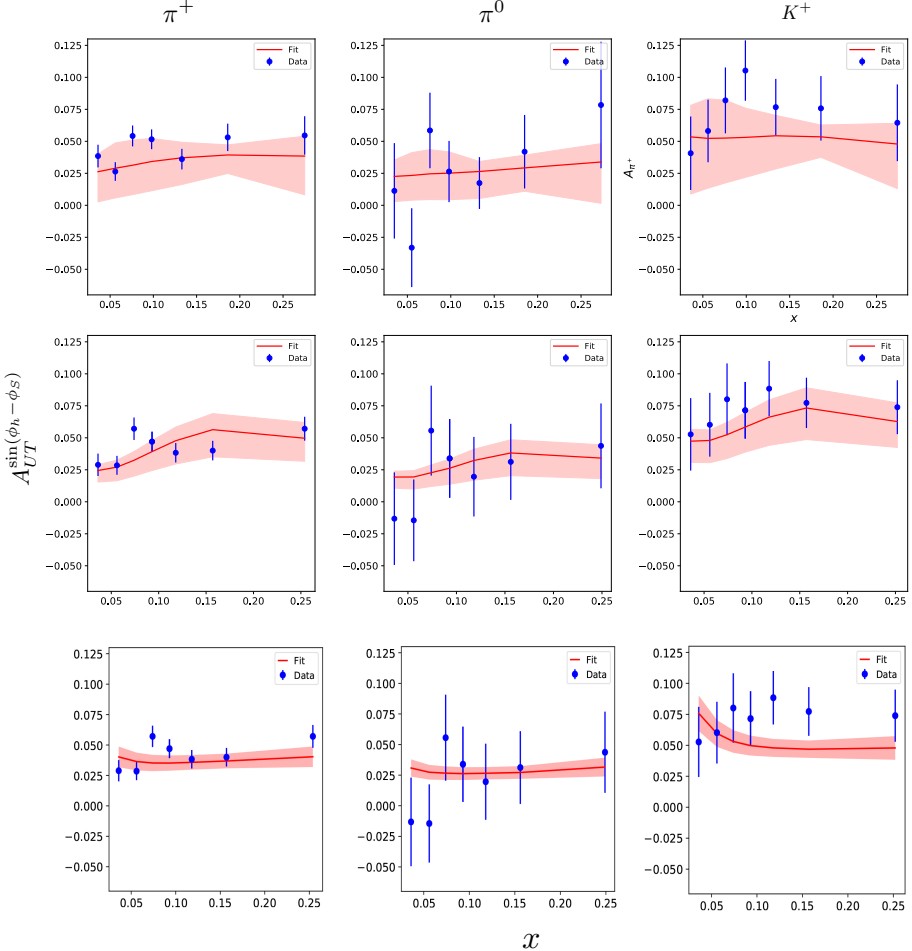

Figure 4: Sivers asymmetry fit results (at $Q^2 = 2.4$ GeV$^2$) of $\pi^+, \pi^0, K^+$ (columns) for HERMES2009(first row), HERMES2020 (second row), and the projected asymmetry values for HERMES2020 kinematics from the Neural Net model trained on HERMES2009 data (third row).

Table 2: Fit results ($\chi^2/ndata$ values) for HERMES2009 and HERMES2020

| Hadron | Dependence | ndata | $\chi^2/ndata$ HERMES2009 | $\chi^2/ndata$(NN) | ndata | $\chi^2/ndata$ HERMES2020 | $\chi^2/ndata$(NN) |
|---|---|---|---|---|---|---|---|
| $\pi^+$ | $x$ | 7 | 2.53 | 2.29 | 8 | 2.12 | 2.23 |
| $\pi^+$ | $z$ | 7 | 1.02 | 1.01 | 11 | 1.49 | 1.63 |
| $\pi^+$ | $p_{hT}$ | 7 | 5.23 | 3.40 | 8 | 1.14 | 2.07 |
| $\pi^-$ | $x$ | 7 | 1.94 | 3.13 | 8 | 1.81 | 2.82 |
| $\pi^-$ | $z$ | 7 | 2.45 | 0.52 | 11 | 1.16 | 0.57 |
| $\pi^-$ | $p_{hT}$ | 7 | 1.61 | 1.96 | 8 | 1.20 | 1.44 |
| $\pi^0$ | $x$ | 7 | 0.85 | 0.90 | 8 | 0.40 | 0.50 |
| $\pi^0$ | $z$ | 7 | 1.11 | 1.13 | 11 | 0.95 | 0.97 |
| $\pi^0$ | $p_{hT}$ | 7 | 2.00 | 1.61 | 8 | 0.50 | 0.73 |
| $K^+$ | $x$ | 7 | 1.22 | 1.78 | 8 | 0.48 | 1.45 |
| $K^+$ | $z$ | 7 | 2.97 | 3.69 | 11 | 6.31 | 7.99 |
| $K^+$ | $p_{hT}$ | 7 | 2.65 | 1.29 | 8 | 1.26 | 2.45 |
| $K^-$ | $x$ | 7 | 0.49 | 0.52 | 8 | 0.26 | 0.54 |
| $K^-$ | $z$ | 7 | 0.52 | 0.57 | 10 | 0.93 | 1.11 |
| $K^-$ | $p_{hT}$ | 7 | 0.96 | 0.73 | 8 | 0.79 | 2.93 |
| Total | | 105 | 1.84 | 1.64 | 134 | 1.477 | 2.02 |

## 5 Conclusions & Future work

The fit results to HERMES2009 data using *iminuit* & neural net model are consistent, therefore the NN trained model was used to generate the Sivers asymmetries for HERMES2020 Kinematics and compared with actual HERMES2020 data. It was observed that the inclusion of the strange quark contribution not only facilitates the fits but also describes a consistent behavior of the Sivers function. It is important to note that the HERMES2020 results from the neural network serve as a test set, as the model was trained only on HERMES2009 data. Such results could indicate over-fitting the HERMES2009 data which could be resolved in future work by incorporating further data sets. Performing the global fits with HERMES, COMPASS (SIDIS data) is currently ongoing and will be published. The results indicate that the neural network representation of the quark contribution offers a promising alternative approach towards model-independent nature and could provide a path toward reducing uncertainties in the Sivers functions because there is a clear need for data and constraining using the Drell-Yan process, and for both Sivers and Boer-Mülders functions.

## 6 Acknowledgements

This work was supported by the Department of Energy (DOE), United States of America contract DE-FG02-96ER40950.

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
