# Peer review of "Sivers extraction with Neural Network"

_SciPost Physics Proceedings, doi:SciPost Phys. Proc. 8, 035 (2022)_

## Round 1 · Referee Report · Anonymous (Referee 1) · 2022-3-1

Report
Their discussion of the fit results to HERMES2009 and HERMES2020 data states they are consistent, but the table of values shows parameters that are substantially incompatible. I suspect they mean to show that their neural network approach matches the observables despite the very different parameters for these two datasets. This may be something understood to a community expert, but is misleading to outsiders. A brief discussion of the two datasets, in particular what differs between them, and how the NN is expected to handle these differences, would be very helpful. The Journal's acceptance criteria for these proceedings are otherwise met.
If a revised manuscript is submitted, please also correct typos where found, eg the first word of the abstract (psuedo --> pseudo) and second-to-last word of the conclusions (Mlders --> Mulders).
If a revised manuscript is submitted, please also correct typos where found, eg the first word of the abstract (psuedo --> pseudo) and second-to-last word of the conclusions (Mlders --> Mulders).

---

## Round 2 · Author Response

Dear Editor,
Thanks for your comments, corrections, and suggestions. The revised manuscript is attached.
Thank you.
Best Regards,
Ishara

---

## Editorial Decision

published